# Expanded Reactor Engineering Calculations for the Oxidative Coupling of Methane

**Andrin Molla, Sonya Rivera, Phillip Pera, Michael Landaverde and Robert Barat ***

Otto H. York Department of Chemical and Materials Engineering, New Jersey Institute of Technology, Newark, NJ 07102, USA; am492@njit.edu (A.M.); sr367@njit.edu (S.R.); pp523@njit.edu (P.P.); msl27@njit.edu (M.L.)
**\*** Correspondence: barat@njit.edu

**Abstract:** The catalytic activation of $CH_4$ by limited amounts of $O_2$ produces a mixture of synthesis gas (CO, $H_2$) and light hydrocarbons ($C_2H_x$), the relative amounts of each depending on catalyst type and process conditions. Using an elementary reaction mechanism for the oxidative coupling of methane (OCM) on a $La_2O_3/CeO_2$ catalyst derived from the literature, this study replaces the activating $O_2$ with moist $H_2O_2$ vapor to reduce synthesis gas production while improving $C_2H_x$ yields and selectivities. As the $H_2O_2$ content of the activating oxidant rises, more of the $CH_4$ conversion occurs in the gas phase instead of with the catalytic surface. In a packed bed reactor (PBR), the use of $H_2O_2$ allows the PBR "light-off" to occur using a lower feed temperature. In exchange for a small decline in $CH_4$ conversion, $C_2H_x$ selectivity increases while synthesis gas production drops. In a continuous stirred tank reactor (CSTR), $H_2O_2$ improves $C_2H_x$ over synthesis gas across a wider range of feed temperatures than is possible with the PBR. This suggests the CSTR will likely reduce OCM preheating requirements.

**Keywords:** catalyst; coupling; methane; peroxide; oxidative

## 1. Introduction

The expanded use of hydraulic fracturing has resulted in the venting and flaring of large volumes of hydrocarbons, especially natural gas. A lack of local pipeline capacity results in more flaring [1] and fugitive emissions. Public sentiment is prompting government environmental regulators to force the reduction or outright banning of hydrocarbon flaring and fugitive emissions [2] to reduce climate change by global warming. Petroleum and natural gas companies are now actively promoting their efforts to reduce their methane footprints [3]. However, the engineering challenge of $CH_4$ conversion is considerable. Catalytic methods offer several conversion approaches.

Catalytic activation of $CH_4$ is generally classified as indirect or direct. Indirect activation produces synthesis gas (primarily CO and $H_2$) using an oxygen source by reforming ($H_2O$—steam; $CO_2$—dry) or partial oxidation ($O_2$). Synthesis gas can be catalytically converted to useful products such as alcohol (usually $CH_3OH$) or higher hydrocarbons (by Fischer–Tropsch process).

Direct activation of $CH_4$ uses no oxygen source. It directly breaks the very strong $CH_3$–H bond ($4.39 \times 10^5$ J/mol). For example, methane dehydroaromatization (MDA) uses a Mo/HZSM-5 zeolite [4] catalyst to form $C_2H_4$ and aromatics at 950–1030 K. Unfortunately, MDA is thermodynamically limited. In addition, catalyst activity drops quickly due to coke deposition.

An intermediate direct approach is the oxidative coupling of methane (OCM) that uses a very small amount of $O_2$ to activate the $CH_4$ while limiting the coke formation. The OCM catalysts are transition metal oxides on an oxide support, e.g., $La_2O_3/CaO$ [5] and $La_2O_3/CeO_2$ [6]. Feed $CH_4/O_2$ molar ratios of 7–11 with temperatures ~840–1220 K

have been studied. The products include $C_{2+}H_x$ and synthesis gas, with the distribution depending on the catalyst and temperature. More feed $O_2$ favors $CH_4$ conversion, but lowers $C_{2+}H_x$ selectivity in favor of $CO_x$.

Gambo et al. [7] reviewed recent advances in OCM, including the use of catalyst nanowires, and identified avenues for further research. Using proprietary nanowire catalysts, Siluria Technologies demonstrated OCM in a continuous flow large demonstration plant [8,9]. The primary goal is the production of $C_2H_4$ for subsequent conversion to gasoline and chemicals. Siluria envisions a flexible two-stage OCM reactor in which the first stage is a packed bed reactor (PBR) feeding $CH_4$, $O_2$, and possibly $C_2H_6$. The second stage feeds more $C_2H_6$ in an endothermic pyrolysis plug flow reactor that uses the first stage exothermicity.

Considering the potential and the constraints of OCM, the reactor type becomes important. Conventional packed beds are not economically viable for OCM [10]. Other configurations such as membrane reactors and fluidized beds should be considered. A recent study [11] compared the packed bed reactor (PBR) and continuous stirred tank reactor (CSTR) for OCM based on calculations with a detailed reaction mechanism [6]. Higher feed temperatures were required to achieve a "light off" of the PBR, while the CSTR required considerably lower feed temperatures to reach nearly comparable conversions. The CSTR—a fluidized bed in likely practice—favored synthesis gas production over $C_{2+}H_x$ as compared to the PBR.

In a difficult experimental study, Liu et al. [12] considered the rapid PBR "light off". Using careful temperature control and real-time product measurement in a micro-reactor during OCM over $La_2O_3$ nanorod catalysts with a feed $CH_4/O_2 = 3$, Liu et al. observed that $CO_2$ is the dominant product at the lower temperatures (<853 K) when relative $O_2$ concentrations are high. The system transitions through a window (~873 K) to higher temperatures (>913 K) that favor $C_{2+}H_x$ at the lower $O_2$ levels. In effect, a competition between $CO_x$ and $C_{2+}H_x$ formation occurs in this window. This suggests that reactor configuration and temperature will be critical for OCM reactor design. These observations also show that, in OCM, the production of byproduct syngas is unavoidable if $C_{2+}H_x$ is the goal.

In typical OCM, some of the limited $O_2$ dissociates on the catalyst surface [13]. An adsorbed $\bullet O_s$ then abstracts an $H\bullet$ from $CH_4$ to form the key $\bullet CH_3$ gas phase radical. Oxygen atoms on the metal oxide lattice surface might directly abstract the $H\bullet$ from $CH_4$. In this case, the gas phase $O_2$ replenishes the resulting surface vacancy, leaving behind an adsorbed $\bullet O_s$ [7]. In either case, the $\bullet CH_3$ radicals can combine to form $C_2H_6$. Further reactions form the desired $C_{2+}H_x$ products, and the undesired $CO_x$ and coke. An alternate $CH_3$-H bond activator would reduce $CO_x$ while possibly enhancing $C_{2+}H_x$ formation. Unfortunately, the desired $C_{2+}H_x$ products are more susceptible to oxidation than the reactant $CH_4$. This can occur via gas phase $O_2$ or surface oxygen species [7,13].

An alternate or supplementary oxidant that might be less aggressive toward $C_{2+}H_x$ by reducing surface oxygen species while still activating the $CH_4$ is the $\bullet OH$ gas phase radical. Gas phase hydroxyl radicals can be formed from the gas phase decomposition of co-fed $H_2O_2$ vapor: $H_2O_2 + M = 2 \bullet OH + M$. Section 3 below summarizes experimental literature on the use of $H_2O_2$ for OCM that motivates this paper.

This paper is also a sequel to the Rivera et al. [11] study on OCM. It uses the same elementary reaction mechanism as that developed by Karakaya et al. [6]. The paper considers vapor phase $H_2O_2$ as a supplemental or alternative activating oxidizer to reduce syngas production in favor of $C_{2+}H_x$. Both CSTR and PBR are considered.

## 2. Kinetic Mechanism and Computational Tool

The detailed reaction mechanism used in this study was developed by Karakaya et al. [6]. It is composed of series and parallel elementary reactions [13] in both the gas and surface phases. The gas phase portion is taken from Chen et al. [14]. The surface portion is inspired by the work of Alexiadis et al. [15].

In the current study, the OCM mechanism was employed in reactor simulations by Detchem® [16]. This program achieves material and energy balances using the mechanism, based on required reactor input data and parameters. In this study, separate adiabatic PBR (modeled as plug flow) and CSTR (modeled as perfectly mixed) runs were conducted with the Detchem® PBED and CSTR applications, respectively. See [11] for a listing of the governing balance equations used in each reactor simulation.

Comparative results were prepared in terms of conversions $X_{CH4}$, selectivities $S_j$ of useful products ($C_2H_x$, CO, $H_2$) or byproducts ($H_2O$, $CO_2$), and yields $Y_j$:

$$X_{CH_4} \equiv \frac{F_{CH_4,in} - F_{CH_4}}{F_{CH_4,in}} \quad X_{O_2} \equiv \frac{F_{O_2,in} - F_{O_2}}{F_{O_2,in}} \quad S_j \equiv \frac{n_j F_j}{F_{CH_4,in} - F_{CH_4}} \quad Y_j \equiv \frac{n_j F_j}{F_{CH_4,in}} \quad (1)$$

where $F_j$ = molar flow rates, $F_{j,in}$ = molar rate at the reactor inlet, and $n_j$ is the number of $CH_4$ moles needed to make one mole of product (byproduct). For example, for $C_2H_4$, $n_j = 2$; for $H_2$, $n_j = 0.5$.

In the prior study [11] of OCM with $O_2$ (no $H_2O_2$) as the activator, twenty cases were considered in separate CSTR and PBR calculations. The process parameters considered are summarized in Table 1. The parametric study included the molar feed $CH_4/O_2$ ratio ("low" = 7, "high" = 11; LR and HR, respectively), molar feed rate ("low" = $8.984 \times 10^{-5}$, "high" = $1.412 \times 10^{-4}$ mole/s; called LF and HF, respectively), and feed temperature (843–1243). At each of the five feed temperatures, four cases were considered: LR_LF, LR_HF, HR_LF, and HR_HF. These conditions were inspired by published laboratory data [6]. Each reactor simulation assumed the same catalytic site density and total catalytic surface area, resulting in the same $2.42 \times 10^{-6}$ total moles of sites. A processing rate can be defined as the ratio of total molar feed rate to the total number of catalyst sites. The processing rate range is 37.1–58.3 s$^{-1}$. These parameters motivate the present study. Larger reactors can be scaled from these conditions.

**Table 1.** Process parameters of this study, inspired by Karakaya et al. [6].

| Low Feed Ratio "LR" $CH_4/O_2$ or $CH_4/H_2O_2 = 7$ | | High Feed Ratio "HR" $CH_4/O_2$ or $CH_4/H_2O_2 = 11$ | | |
|---|---|---|---|---|
| Low Feed Processing Rate "LF" 37.1/s | | High Feed Processing Rate "HF" 58.3/s | | |
| **Feed Temperatures $T_{in}$ (K)** | | | | |
| 843 | 943 | 1043 | 1143 | 1243 |

## 3. Alternate Activator $H_2O_2$

Garibyan et al. [17] studied OCM over Pb/aerosil, ZnO, and 10% $Na_2O$/ZnO catalysts. Pulses of $H_2O_2$ vapor into the $CH_4/O_2$ feed increased the $C_2H_x$ yield while stabilizing catalytic activity during OCM at 1 atmosphere and 1023 K. With a 1% Au/5% $La_2O_3$/CaO catalyst, at 973–1073 K, Eskendirov et al. [18,19] observed that $H_2O_2$ increased $CH_4$ conversion, while enhancing $C_{2+}$ hydrocarbon yields even up to benzene. They speculated that $H_2O_2$ decomposition resulted in more •OH radicals for gas phase activation of the $CH_4$. They even observed OCM in the presence of $H_2O_2$ vapor without catalyst at temperatures as low as 673 K, with considerable selectivity for $C_2H_x$. These studies motivate this paper.

Considering the importance of making any OCM process as "green" as possible, a potentially sustainable source of $H_2O_2$ uses a photo-activated $TiO_2$-Au-Si catalyst while feeding $O_2$ and liquid $H_2O$ [20]. Spiegelman and Alvarez [21] developed a simple yet clever technology to produce a continuous vapor stream of $H_2O_2$ from a liquid solution of $H_2O_2$ in water. Subsequent drying of the vapor stream to raise the $H_2O_2$ concentration runs the risk of energetic decomposition, thus posing a safety risk. In a study of the decomposition of $H_2O_2$ vapor on various surfaces, Satterfield and Stein [22] generated $H_2O_2$ vapor concentrations of up to 0.23 atm in a 1 atm system. Therefore, in the remainder

of these calculations, we used a conservative molar $H_2O/H_2O_2 = 4$ linkage in all cases where $H_2O_2$ was used.

The OCM process requires considerable preheating, so the $H_2O_2$ decomposition risk also calls into question the feed temperature for the $H_2O/H_2O_2$ vapor stream. Consider the decomposition: $H_2O_2 + M \rightarrow 2$ •OH + M with a rate constant borrowed from the Chen et al. [14] reaction set for the $CH_4$ gas phase chemistry used in the OCM mechanism [6]. Assume a feed $CH_4/H_2O_2$ ratio of 11 (the HR case), with the coflowing $H_2O$ vapor, and no $O_2$. A simple kinetic calculation shows that, at 1243 K, the $H_2O_2$ will be 100% decomposed in 10 microseconds. At 673 K, the time is a more realistic 5 s. This simple calculation suggests that preheating a combined $CH_4$, $O_2$, $H_2O_2$, and $H_2O$ feed stream would be problematic, especially for a PBR. It also suggests keeping the $CH_4/O_2$ and $H_2O/H_2O_2$ vapor streams separate, with the $CH_4/O_2$ stream taking most or all the preheat. Separate feed streams are more easily handled with the CSTR.

Finally, the replacement of $O_2$ by $H_2O_2$ maintains the feed $CH_4$-to-O molar ratio, though it somewhat increases the overall H content of the feed. Consider the following overall reactions below. Though simplistic, Equations (1) and (2) show that replacing $O_2$ with $H_2O_2$ should increase the production of $C_2H_4$ and $H_2O$, while reducing CO and $H_2$. In addition, adiabatic reactor temperatures should be lower.

$$CH_4 + 0.5\, O_2 = \frac{1}{3}C_2H_4 + \frac{2}{3}H_2O + \frac{1}{3}CO + \frac{2}{3}H_2 \quad \Delta H_r^o = -105.7 \text{ kJ/mole} \quad (2)$$

$$CH_4 + 0.5\, H_2O_2 = 0.5\, C_2H_4 + H_2O + 0.5\, H_2 \quad \Delta H_r^o = -72.7 \text{ kJ/mole} \quad (3)$$

### 3.1. Incremental $H_2O_2$ Replacing $O_2$ to PBR at Fixed Feed Temperature

Rivera et al. [11] showed that, for the parametric range considered (Table 1), the highest $CH_4$ conversions for the PBR were nearly 40% for the LR cases, with little impact of flow rate, for a 1243 K feed temperature. The LR_LF case showed the highest sum_C2Hx (sum of $C_2H_6$, $C_2H_4$, and $C_2H_2$) selectivities and yields. First, for the same 1243 K feed temperature, $H_2O_2$ was incrementally substituted for $O_2$. It was assumed that $H_2O_2$ vapor will be available at 20 mole percent with the balance of $H_2O$ vapor. The starting point was the LR_LF (feed $CH_4/O_2 = 7$, feed processing rate 37.1/s) case [11] in the PBR. The PBR bed length was that used in the experiments described elsewhere [6]. This high feed temperature did ignore the $H_2O_2$ stability issue. The insights gained, however, will help identify the utility of $H_2O_2$ as a potential $CH_4$ activator.

Figure 1 shows that the final PBR bed temperature drops with increasing $H_2O_2$ content. This was attributed to the reduced overall reaction exothermicity suggested by Equations (1) and (2). However, the bed temperature peaked much earlier when $H_2O_2$ was used in the feed oxidant. A closer examination of the post-entrance region is revealed in Figure 2 below. The $H_2O_2$-containing cases showed a near-immediate rise from the 1243 K feed temperature. The very early single peak for the R = 1 (no feed $O_2$) case roughly corresponded to the exhaustion of the $H_2O_2$, which was consistent with the extremely rapid $H_2O_2$ decomposition described earlier at this temperature. The presence of the feed $O_2$ caused a second local temperature peak further downstream. These peaks (R = 0.33, 0.67 cases) and the much later single peak (R = 0) corresponded approximately to where the $O_2$ ran out, with no further adsorbed •$O_s$.

The R = 1 case (no feed $O_2$) showed almost no adsorbed species, suggesting that all the $CH_4$ conversion effectively occurred in the gas phase (i.e., non-catalytic). On the contrary, for the R = 0.33 and 0.67 cases, while $H_2O_2$ was still present, there was a complex parallel/series scheme ongoing with both catalytic and gas-phase reactions occurring. These observations were consistent with the claim that the $CH_4$ conversion is accelerated by •OH gas phase radicals produced from the $H_2O_2$ dissociation [18,19].

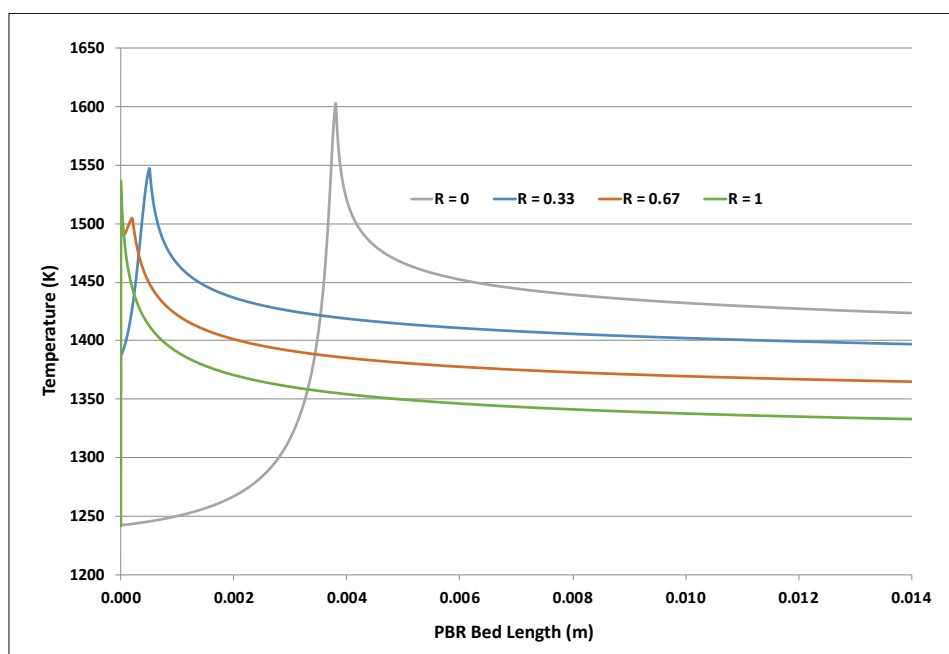

**Figure 1.** Impact of $H_2O_2$ content in feed oxidant; PBR LR_LF case at 1243 K feed; curves are different feed molar values $R \equiv H_2O_2/(O_2 + H_2O_2)$.

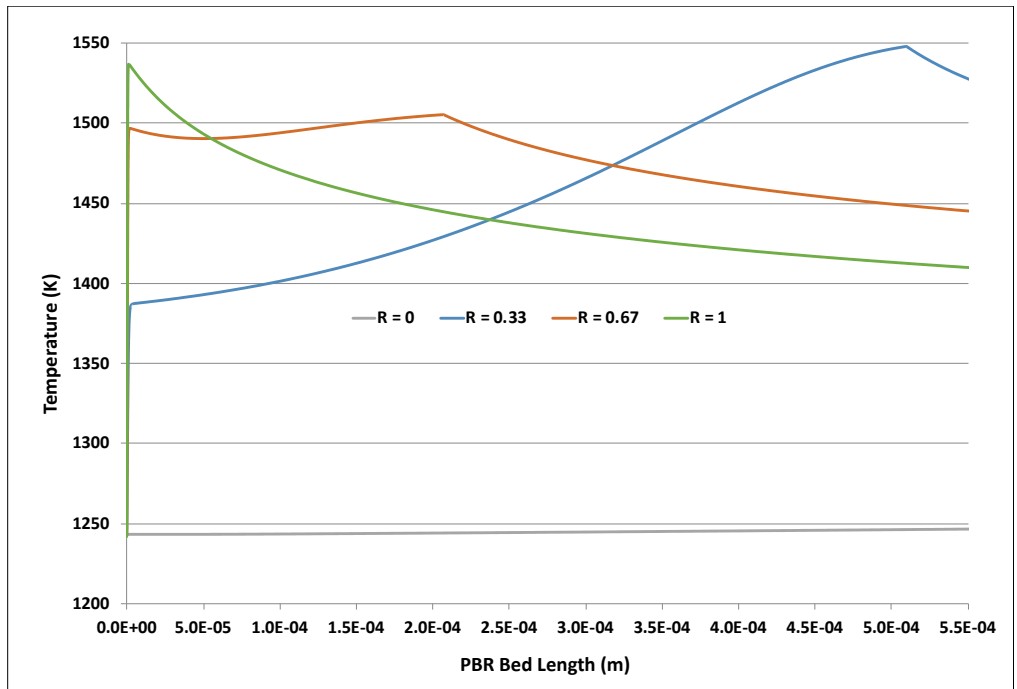

**Figure 2.** Post-entrance region of PBR for LR_LF case at 1243 K feed; curves are different feed molar values $R \equiv H_2O_2/(O_2 + H_2O_2)$.

Figure 3 shows that increasing the $H_2O_2$ content improved the selectivity of sum_$C_2H_x$, while lowering both the selectivity and yield for syngas ($H_2$ + CO). There was a negligible impact on sum_$C_2H_x$ yields. The reduction in syngas was due almost entirely to a reduction in CO. Finally, for these four cases from R = 0 to 1, the $CH_4$ conversions were: 39.3, 36.6, 35.1, and 33.8%, respectively. In all cases, the final $O_2$ and $H_2O_2$ conversions were 100%.

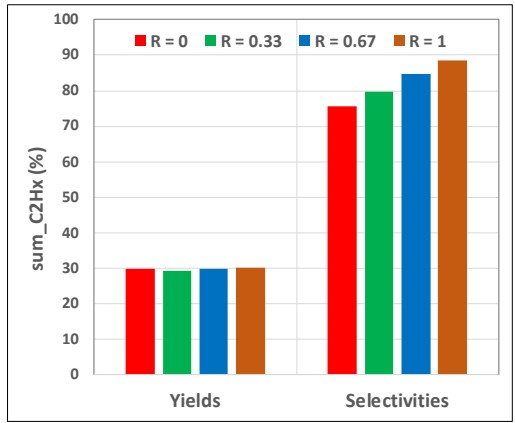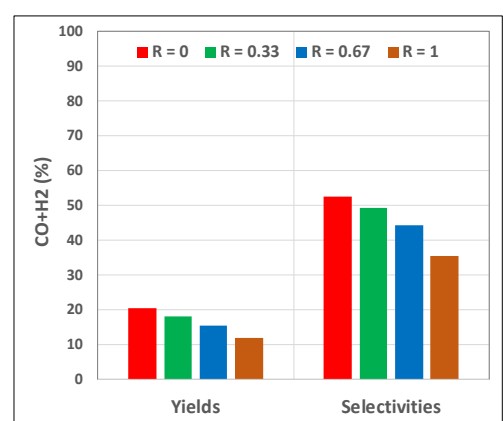

**Figure 3.** Impact of $H_2O_2$ content in feed oxidant; PBR LR_LF case at 1243 K feed; $R \equiv H_2O_2/(O_2 + H_2O_2)$. Values are based on PBR outlet; (**left**) sum$C_2H_x = C_2H_6 + C_2H_4 + C_2H_2$; (**right**) $CO + H_2$.

An expanded look at the long-post-entrance region provides more insight into the dramatic impact of substituting some of the feed $O_2$ with $H_2O_2$ vapor. Figure 4 shows the selectivities for the CO, $C_2H_6$, $C_2H_4$, and $CH_4$ conversions for the LR_LF case, at 1243 K feed temperature, for feed ratio cases R = 0 and R = 0.33. The curves were almost unchanged after the 0.006 m bed length was reached.

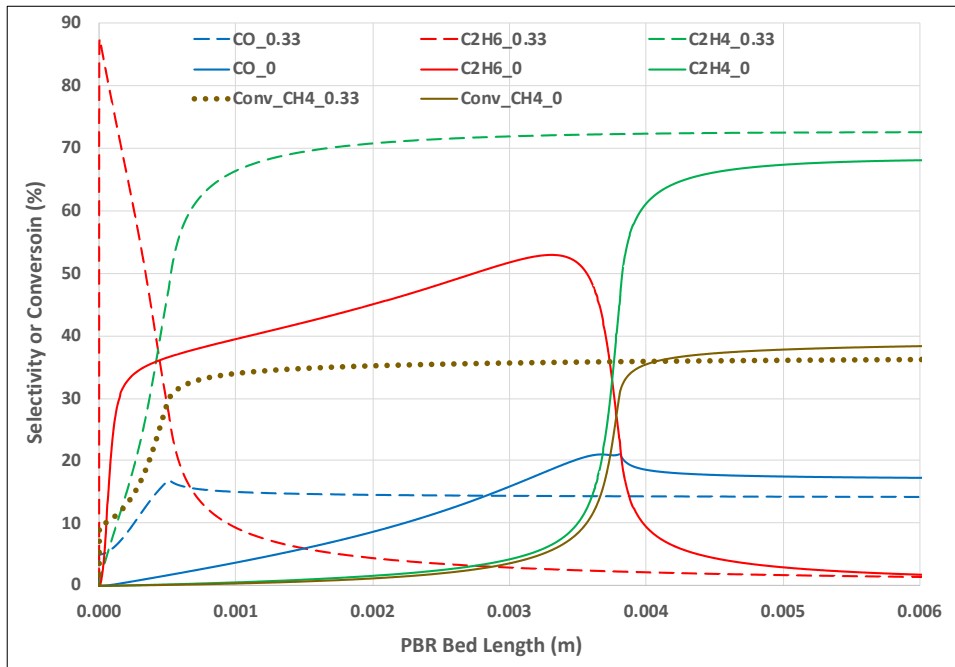

**Figure 4.** PBR long-post-entrance region for LR_LF case, 1243 K feed; key sensitivities and conversions for R = 0, 0.33 oxidant ratios.

Some key points can be made here. In the R = 0 case, CO selectivity exceeded $C_2H_4$ before the temperature peaked (see Figure 1), but was lower than $C_2H_6$. After the peak temperature, consistent with experimental observations by Liu et al. [12], $C_2H_4$ exceeded CO. In the R = 0.33 case, the $H_2O_2$ (not shown) dissociated immediately upon entry. The resulting •OH radicals abstracted •H atoms from $CH_4$, causing a spike in $C_2H_6$ formation, and a quickly rising $CH_4$ conversion. The $C_2H_6$ rapidly dehydrogenated to $C_2H_4$. The CO peaked at approximately where the temperature peaked. Unlike the R = 0 case, both $C_2$ species exceeded CO prior to the temperature peak. This all occurred much faster than

for the R = 0 case. The ultimate $CH_4$ conversion for the R = 0.33 case was only slightly lower than for the R = 0 case, while showing a higher $C_2H_4$ selectivity and lower CO. Liu et al. [12], for the R = 0 case, concluded that the selectivities of $CO_x$ and $C_2$ depended on local $O_2$ concentration and temperature. Using $H_2O_2$ added the further complexity of gas phase chemistry to the surface reactions.

### 3.2. Use of $H_2O_2$ to Decrease Feed Temperature to PBR

We now discuss whether the replacement of $O_2$ by $H_2O_2$ allows for a lowering of the overall PBR feed temperature, which would be an energy and cost saving. This analysis used the R = 0 and R = 0.33 feeds with the LR_LF case, with the results shown in Table 2.

**Table 2.** Impact of partial replacement of feed $O_2$ by $H_2O_2$ at various feed temperatures for PBR running LR_LF case.

| Feed Temperature (K) | 1243 | 1143 | 1043 | 1043 | 943 | 943 |
|---|---|---|---|---|---|---|
| R value | 0 | 0 | 0 | 0.33 | 0 | 0.33 |
| Max Temp (K) | 1603 | 1547 | 1065 | 1414 | 944 | 1345 |
| Exit Temp (K) | 1424 | 1423 | 1065 | 1343 | 944 | 1321 |
| $CH_4$ Conv (%) | 39.3 | 33.8 | 0.891 | 26.4 | 0.044 | 21.8 |
| sum_$C_2H_x$ Selec (%) | 75.6 | 72.3 | 58.7 | 70.0 | 36.7 | 62.6 |
| sum_$C_2H_x$ Yield (%) | 29.7 | 24.4 | 0.522 | 18.5 | 0.016 | 13.6 |
| CO + $H_2$ Sel. (%) | 52.4 | 51.6 | 24.3 | 54.5 | 10.2 | 57.8 |
| CO + $H_2$ Yield (%) | 20.6 | 17.5 | 0.217 | 14.4 | 0.004 | 12.6 |

The partial $H_2O_2$ substitution for $O_2$ produced a respectable $CH_4$ conversion at the lower feed temperatures where $O_2$ feed alone showed no OCM activity (R = 0 for 1043 and 943 K feeds). These results at lower feed temperatures were consistent with those observed experimentally [17–19]. At the 843 K feed temperature, even the R = 0.33 case was poor.

### 3.3. Incremental $H_2O_2$ Replacing $O_2$ to CSTR

As mentioned above, the $CH_4/O_2$ and $H_2O/H_2O_2$ vapor streams would likely be fed separately into the CSTR due to safety concerns about preheating a vapor stream containing $H_2O_2$. For example, for the R = 0.33 and HR_HF case, to achieve an effective (hypothetical) 843 K feed temperature while holding the $H_2O/H_2O_2$ stream at 373 K, the $CH_4/O_2$ stream would be preheated to about 883 K. The HR_HF case was chosen for this CSTR analysis because it showed the best yield and selectivity of sum_$C_2H_x$ at the lowest feed temperatures in the earlier study [11].

For the CSTR calculations, the volume was the same as the open (gas) volume of the packed bed, with the same catalyst surface area. The CSTR might be a single-phase ideal fluidized bed, or a perfectly mixed (e.g., jet-stirred) reactor with catalyst on the walls.

Substitution of all or some of the feed $O_2$ content with $H_2O_2$ had a marked impact on the CSTR performance. Figure 5 (left) shows that substituting $H_2O_2$ for $O_2$ reduced the exit temperature somewhat (~65–110 K), as might be expected from the lower exothermicity (see Equations (2) and (3)). However, Figure 5 (right) shows a complex story for the impact on $CH_4$ conversion. For effective feed temperatures of 843 and 943 K, switching from $O_2$ to $H_2O_2$ reduced $CH_4$ conversion by only ~3 percentage points. With a 1043 K feed temperature, there was little impact on conversion. At 1143 and 1243 K, switching to $H_2O_2$ actually increased $CH_4$ conversion. While literature experiments [17–19] used a PBR, the results here were still found to be consistent with those observations in terms of the activity of $H_2O_2$.

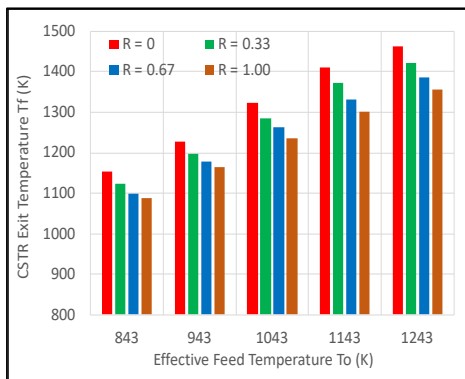 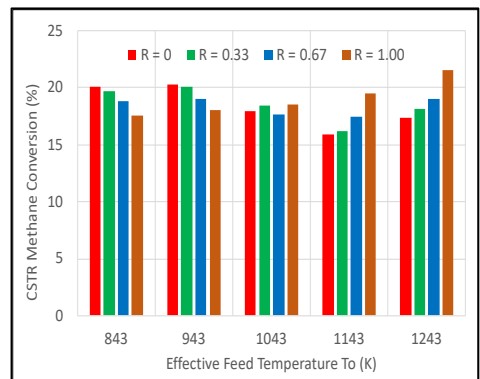

**Figure 5.** Impact of $H_2O_2$ content in feed oxidant on CSTR exit temperature (**left**) and $CH_4$ conversion (**right**) for HR_HF case where molar R = $H_2O_2/(O_2 + H_2O_2)$.

Figure 6 shows that the yield of sum_$C_2H_x$ was notably higher than the CO + $H_2$ yield for all R cases, while increasing $H_2O_2$ had a greater impact on yields at the higher feed temperatures. These results were consistent with those revealed in Figure 3 for the PBR. Selectivities of sum_$C_2H_x$ remained higher than CO + $H_2$, especially at the lower feed temperatures, as seen in Figure 7.

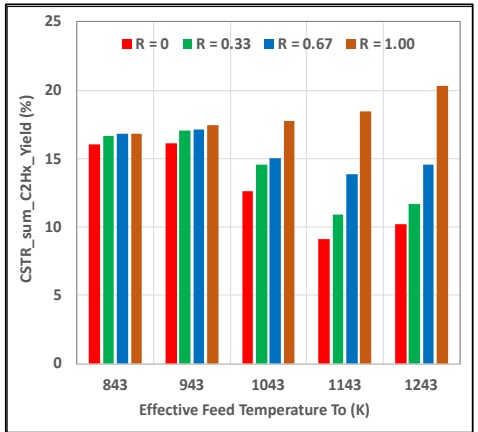 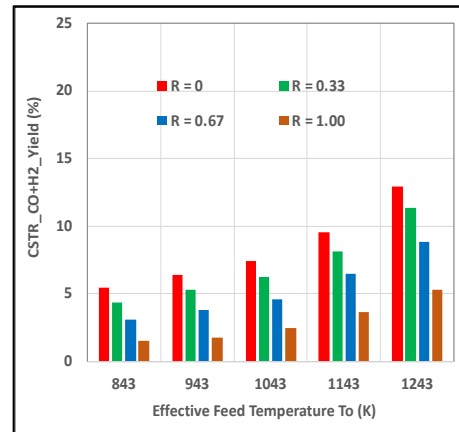

**Figure 6.** Impact of $H_2O_2$ content in feed oxidant on sum_$C_2H_x$ and CO + $H_2$ yields in CSTR for HR_HF case where molar R = $H_2O_2/(O_2 + H_2O_2)$; (**left**) sum$C_2H_x$ = $C_2H_6 + C_2H_4 + C_2H_2$; (**right**) CO + $H_2$.

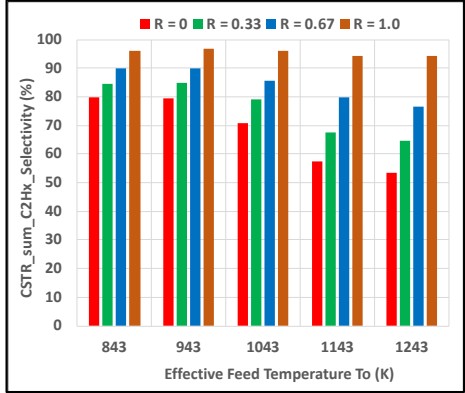 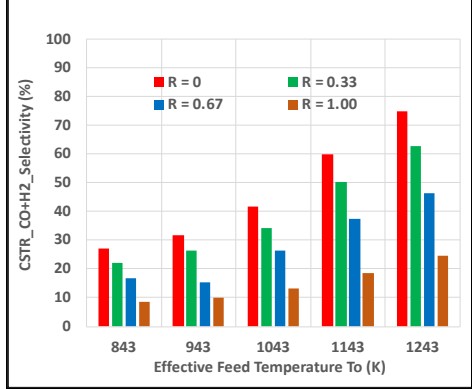

**Figure 7.** Impact of $H_2O_2$ content in feed oxidant on sum_$C_2H_x$ and CO + $H_2$ selectivities in CSTR for HR_HF case where molar R = $H_2O_2/(O_2 + H_2O_2)$; (**left**) sum$C_2H_x$ = $C_2H_6 + C_2H_4 + C_2H_2$; (**right**) CO + $H_2$.

Equations (2) and (3) suggest that replacing $O_2$ by $H_2O_2$ will increase the production of $C_2H_4$ and $H_2O$, while reducing CO and $H_2$. Consider the HR_HF case, with the effective feed temperature into the CSTR of 1243 K, with the results shown in Table 3. As the $H_2O_2$ fraction in the oxidant increased (i.e., higher R value), the CSTR exit temperature fell, but the $CH_4$ conversion increased. The $C_2H_4$ and $H_2O$ production increased, while CO and $H_2$ dropped. Finally, the fraction of catalytic sites occupied by adsorbed O atoms ($O_s$) decreased as R increased. Since ●H abstraction by ●$O_s$ is the primary catalytic step for $CH_4$ activation [6] by $O_2$, the drop in ●$O_s$ fraction was consistent with a shift from heterogeneous catalyzed to homogeneous non-catalyzed conversion pathways at higher R.

**Table 3.** Impact of progressive replacement of feed $O_2$ by $H_2O_2$ at 1243 K effective feed temperatures for CSTR running HR_HF case. * $H_2O$ values are corrected for $H_2O$ in feed for $R \neq 0$ cases.

| R Value | 0 | 0.33 | 0.67 | 1 |
|---|---|---|---|---|
| Exit Temp (K) | 1463 | 1421 | 1386 | 1356 |
| $CH_4$ Conv (%) | 17.3 | 18.1 | 19.0 | 21.5 |
| $C_2H_4$ Selec (%) | 46.1 | 52.6 | 58.8 | 70.0 |
| $C_2H_4$ Yield (%) | 7.98 | 9.50 | 11.2 | 15.1 |
| $H_2O$ Selec (%) * | 29.1 | 34.3 | 38.5 | 42.1 |
| $H_2O$ Yield (%) * | 5.03 | 6.19 | 7.32 | 9.07 |
| CO Selec (%) | 35.1 | 26.7 | 14.9 | 0.10 |
| CO Yield (%) | 6.05 | 4.82 | 2.84 | 0.02 |
| $H_2$ Selec (%) | 39.8 | 36.1 | 31.5 | 24.5 |
| $H_2$ Yield (%) | 6.89 | 6.51 | 5.99 | 5.27 |
| $O_s$ coverage (ppm) | 24.5 | 15.8 | 9.61 | 0.48 |

### 3.4. Brief Reactor Comparison Summary

A simple comparison between the PBR and CSTR for OCM with and without $H_2O_2$ as an activating oxidant is shown in Table 4. This illustration is based on the LR_LF case (see Table 1). The results will vary somewhat for other cases, but the observations will be similar. This summary considers both the current results and those published earlier with just $O_2$ as an activator [11]. While the claims are based on calculations using the Karakaya et al. [6] mechanism for $La_2O_3/CeO_2$ catalyst, it is anticipated other OCM catalysts would give rise to similar claims.

**Table 4.** Brief comparison of PBR vs. CSTR for OCM using the LR_LF case and R = 0.33 for the $O_2/H_2O_2$ runs.

| | PFR | PFR | CSTR | CSTR |
|---|---|---|---|---|
| | $O_2$ | $O_2/H_2O_2$ | $O_2$ | $O_2/H_2O_2$ |
| Lowest practical feed temperature (K) | 1143 | 943 | 843 | 843 |
| $CH_4$ conversion | 34 | 22 | 23 | 28 |
| Sum_$C_2$Hx selectivity | 72 | 63 | 55 | 80 |
| CO + $H_2$ selectivity | 52 | 58 | 57 | 33 |

Table 4 shows several points. The lowest practical feed temperature is the value below which there is no appreciable $CH_4$ conversion. All remaining values in each column correspond to those temperatures. Replacing a portion of the feed $O_2$ with $H_2O_2$ vapor allows the CSTR to achieve good $CH_4$ conversions at the lowest feed temperature. It also allows the PBR to run with a reduced feed temperature. Even at this low feed temperature, the CSTR has a sum_$C_2H_x$ selectivity that exceeds the PBR at a much higher temperature. The CSTR also shows reduced syngas (CO + $H_2$) and improved sum_$C_2H_x$ selectivity when using the $H_2O_2$.

## 4. Simple Layout of a "Green" OCM Plant

Although Equations (2) and (3) show that OCM via $O_2$ and $H_2O_2$ is exothermic, a future sustainable OCM plant must consider $O_2$ and $H_2O_2$ production and overall plant heat integration. Figure 8 offers a simple schematic. The OCM reactor feeds $CH_4$ and a combination of $O_2$ and vapor phase moist $H_2O_2$. The $O_2$ is produced in a solar powered air separation plant [23] that enjoys the energy and economic savings from chemical looping instead of cryogenic separation [24,25]. Aqueous $H_2O_2$ is produced by the solar powered catalyzed reaction of $O_2$ and acidic liquid $H_2O$ [20]. Vapor phase $H_2O_2$ is stripped out of the liquid by the $N_2$ or He [21] recovered from the natural gas. Heavier-than-$CH_4$ saturated hydrocarbons ($C_2H_6$-$C_5H_{12}$) are separated out from the natural gas. Post-OCM reactor processing separates the CO and $H_2$ as synthesis gas and the desirable coupled hydrocarbons (e.g., $C_2H_4$ and $C_2H_6$). Byproduct $CO_2$ from Separations_2 and Pretreatment can be captured with caustic scrubbing, and subsequently sequestered.

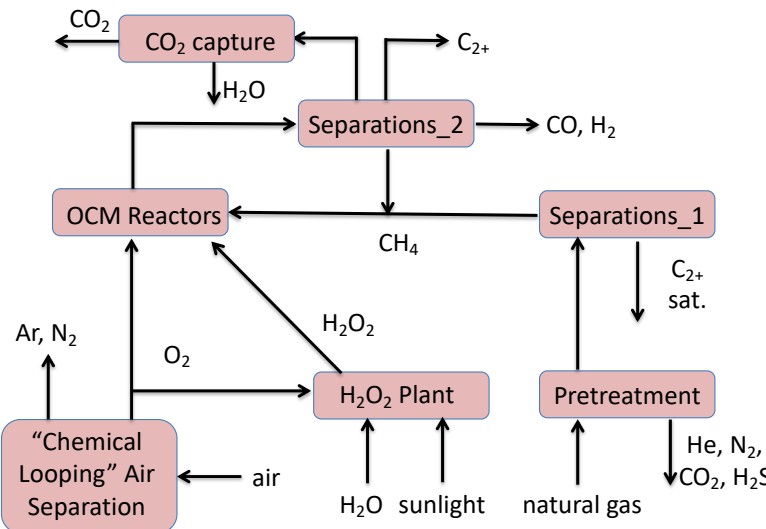

**Figure 8.** Hypothetical "green" OCM plant utilizing solar-powered $H_2O_2$ production and chemical looping for air separation.

## 5. Conclusions

Using an elementary reaction mechanism for the oxidative coupling of methane (OCM) on a $La_2O_3$/$CeO_2$ catalyst borrowed from the literature, this study considered the incremental replacement of the activating $O_2$ with moist $H_2O_2$ vapor. Both packed bed reactor (PBR) and continuous stirred tank reactor (CSTR) configurations were used. As the $H_2O_2$ content of the oxidant increased, more of the $CH_4$ conversion occurred in the gas phase with less assistance from the catalytic surface. Hydroxyl (•OH) radicals from rapid $H_2O_2$ decomposition abstracted •H atoms from $CH_4$ to produce •$CH_3$ radicals. This occurred in parallel to a similar abstraction by oxygen atoms (•$O_s$) adsorbed on the catalyst surface when $O_2$ was fed. In the PBR, $H_2O_2$ allowed the "light-off" temperature jump to occur using a lower feed temperature. Even though there was a slight decline in $CH_4$ conversion, the $C_2H_x$ selectivity increased while synthesis gas dropped. Since significant preheat was still needed, process safety considerations might dictate that $H_2O_2$ vapor is better suited to the continuous stirred tank reactor (CSTR) configuration where the $H_2O_2$/$H_2O$ vapor stream can be fed at lower temperatures separately from the preheated $CH_4$/$O_2$ stream. In a CSTR, $H_2O_2$ significantly improved $C_2H_x$ production compared to synthesis gas over all feed temperatures studied, thus showing that OCM is possible with significantly less preheating compared to PBR. A future OCM plant can operate in a more "green" way with the use of solar-activated $H_2O_2$ production, and solar-powered $O_2$ production from chemical-looping air separation.

**Author Contributions:** A.M.—calculations, S.R.—calculations, P.P.—calculations, M.L.—calculations, R.B.—calculations, writing, supervision, total responsibility. All authors have read and agreed to the published version of the manuscript.

**Funding:** This research received no external funding.

**Institutional Review Board Statement:** Not applicable.

**Informed Consent Statement:** Not applicable.

**Data Availability Statement:** Supporting data are not posted, but are available upon request.

**Acknowledgments:** The authors appreciate the support of Canan Karakaya in using Detchem®.

**Conflicts of Interest:** The authors declare no conflict of interest.

**Sample Availability:** Compound samples are not available from the authors.

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
