# Peer review of "Expanded Reactor Engineering Calculations for the Oxidative Coupling of Methane"

_methane, doi:10.3390/methane1010005_

Round 1

Reviewer 1 Report

This manuscript describes the OCM process with two reactor configuration(PFR and CSTR) using H2O2 as oxidant. It is an interesting topic and can have great impact. However, the current state of manuscript is premature and should be improved before publication.

  1. There is little description of the simulation details. The authors should add the details of mechanism, process parameters, etc.
  2. The authors observed an improved selectivity of C2Hx from simulation. I would expect more in-depth analysis of why. Was it due to kinetic reason? thermodynamic reason? or other factors? The current section is more like a experimental report recording the results.
  3. Section 5 is not closely related to the bulk of manuscript. I'd suggest the authors either delete it or improve it with more details, such as detailed energy analysis/econ analysis etc. 

Reviewer 2 Report

This manuscript fits well with the scope of the journal

I suggest some additions to the manuscript in order to improve it

First of all, a thermodynamic part should be added at least for the H2O2 addition. 

Concerning the 2 reactors presented, I suggest to point out better the diffeerences in terms of limits and conditions used and to present the differencies in a summary in a Table.

The conclusion should also be improved to show the novelty

Round 2

Reviewer 1 Report

The revised manuscript resolved my previous comments and can be published. One minor is the authors should pay attention to significant figures. (ie. the second table 4 vs. the first table 4).